

# MIPhy: identify and quantify rapidly evolving members of large gene families

David M. Curran[1], John S. Gilleard[2] and James D. Wasmuth[1]

[1] Department of Ecosystem and Public Health, Faculty of Veterinary Medicine, University of Calgary, Calgary, AB, Canada
[2] Department of Comparative Biology and Experimental Medicine, Faculty of Veterinary Medicine, University of Calgary, Calgary, AB, Canada

## ABSTRACT

After transitioning to a new environment, species often exhibit rapid phenotypic innovation. One of the fastest mechanisms for this is duplication followed by specialization of existing genes. When this happens to a member of a gene family, it tends to leave a detectable phylogenetic signature of lineage-specific expansions and contractions. These can be identified by analyzing the gene family across several species and identifying patterns of gene duplication and loss that do not correlate with the known relationships between those species. This signature, termed phylogenetic instability, has been previously linked to adaptations that change the way an organism samples and responds to its environment; conversely, low phylogenetic instability has been previously linked to proteins with endogenous functions. With the increase in genome-level data, there is a need to identify and quantify phylogenetic instability. Here, we present Minimizing Instability in Phylogenetics (MIPhy), a tool that solves this problem by quantifying the incongruence of a gene's evolutionary history. The motivation behind MIPhy was to produce a tool to aid in interpreting phylogenetic trees. It can predict which members of a gene family are under adaptive evolution, working only from a gene tree and the relationship between the species under consideration. While it does not conduct any estimation of positive selection—which is the typical indication of adaptive evolution—the results tend to agree. We demonstrate the usefulness of MIPhy by accurately predicting which members of the mammalian cytochrome P450 gene superfamily metabolize xenobiotics and which metabolize endogenous compounds. Our predictions correlate very well with known substrate specificities of the human enzymes. We also analyze the *Caenorhabditis* collagen gene family and use MIPhy to predict genes that produce an observable phenotype when knocked down in *C. elegans*, and show that our predictions correlate well with existing knowledge. The software can be downloaded and installed from https://github.com/dave-the-scientist/miphy and is also available as an online web tool at http://www.miphy.wasmuthlab.org.

Corresponding authors
David M. Curran,
dmcurran@ucalgary.ca
James D. Wasmuth,
jwasmuth@ucalgary.ca

## INTRODUCTION

In the absence of specific selective pressures, the phylogeny of a multi-species gene family will tend to agree with the underlying species tree. However, gene events such as gene duplication/loss, horizontal gene transfer (HGT), and incomplete lineage sorting (ILS)—where a polymorphic locus in an ancestral species results in incongruence with the species tree—may become fixed in a species due to evolutionary processes. These events can result in lineage-specific variations in gene family size and incongruence between the gene family phylogeny and the species tree, properties that have collectively been referred to as "phylogenetic instability" (*Thomas, 2007*). Attempting to work backwards and determine the sequence of events that led from the species tree to the observed gene family is a process called event-inference reconciliation.

It has been hypothesized that the change in environment during a speciation event may lead to higher levels of phylogenetic instability (*Lynch & Conery, 2000*; *Zhang, 2003*; *Hurley, Hale & Prince, 2005*), especially in genes involved in responding to molecules from the environment (xenobiotics). This has been observed in gene families involved in the immune response (*de Bono, Madera & Chothia, 2004*; *Nei, Gu & Sitnikova, 1997*; *Su et al., 1999*), chemosensory receptors (*Niimura & Nei, 2005*; *Thomas et al., 2005*), detoxification (*Thomas, 2007*), and host-pathogen interactions (*Wasmuth et al., 2012*). These observations are supported by recent ecological experimental evidence showing that higher rates of evolution allow populations to more rapidly expand into new territory (*Szűcs et al., 2017*).

Here, we propose using phylogenetic instability to predict the functional roles of the members of a gene family using a new tool, Minimizing Instability in Phylogenetics (MIPhy). Specifically, to identify which family members are under pressure to duplicate and contribute to altered or new functions, with the possibility of new phenotypes. Understanding the effects of these selective pressures is of more than purely theoretical importance; as one example the rapid evolution of drug resistance remains one of the most significant challenges in managing both human (*Saunders & Lon, 2016*) and livestock parasites (*Kaplan & Vidyashankar, 2012*), and the mechanisms underlying these resistant phenotypes is often unknown. We show the usefulness of MIPhy by validating it against two data sets: the cytochrome P450 (*cyp*) genes from ten species of vertebrates, and the collagens from eight species of free-living nematodes. This tool can be used to prioritize genes for further study, for example, by predicting the origin of some species-specific function or identifying essential genes as new therapeutic targets in pathogens. The process to detect phylogenetically unstable genes is twofold. First, a tree of a large multi-member gene family is split into meaningful clusters—termed minimum instability groups (MIGs)—by incorporating an event-inference model of gene evolution. Second, each MIG is independently scored for phylogenetic instability.

### Related work

There are several existing algorithms for species/gene tree reconciliation, but none are able to segregate a gene tree into meaningful clusters, quantify the stability of those gene clusters, or score each gene in order to compare and rank the individual family members.

CAFE 3 uses a stochastic birth-death model of gene family evolution to infer the size of ancestral families (*De Bie et al., 2006*; *Han et al., 2013*). It implements a sampling procedure to determine the statistical significance of those gene families that differ from their expected values, and models the effects of genome assembly and gene annotation errors to provide a more accurate estimate of its evolutionary rates. CAFE 3 uses only the gene family counts without considering the phylogenetic relationships within them, and so would be unable to distinguish inherited paralogs from independently duplicated genes. Further, the algorithm calculates whether an entire gene family is under adaptive evolution, while we are interested in the relative differences between specific clusters of genes within a family. Because of this, it is more suited for large-scale analyses of many gene families at once.

BadiRate is similar to CAFE 3, implementing several additional stochastic models of evolution, and providing three statistical frameworks to calculate significance (*Librado, Vieira & Rozas, 2012*). While it allows for more detailed analyses of species traits, it still relies on gene count data and so is unsuitable here for the same reasons as CAFE. It also requires a species tree with meaningful branch lengths, the creation of which is in itself a challenging analysis.

NOTUNG (*Chen, Durand & Farach-Colton, 2000*; *Vernot et al., 2008*; *Stolzer et al., 2012*) implements a parsimony-based reconciliation algorithm. It finds the sequence of gene events (gene duplication, gene loss, HGT, and ILS) explaining the differences between the observed gene tree and the underlying species relationships that minimizes a weighted sum. Uniquely amongst other reconciliation methods, it allows for the species or gene tree to be non-binary; as the true history of many species is unclear, polytomies can be useful to describe the current state of knowledge. Important in this consideration is that NOTUNG explicitly models HGT and assumes that ILS is a very rare event, only considering it at polytomies in the gene tree. A recent paper has proposed a similar algorithm, with advances in identifying ILS and HGT (*Chan, Ranwez & Scornavacca, 2017*). Identifying HGT is a computationally intensive process and is unlikely to play an important role in gene families from multi-cellular organisms, and we assume that incongruence (as produced by ILS, adaptive evolution, or any other mechanism) is a common enough event to allow throughout the tree (*Carstens & Knowles, 2007*; *Mirarab, Bayzid & Warnow, 2016*; *Scally et al., 2012*). RANGER-DTL is another reconciliation method, and has been reported to be 1,000–1,000,000× faster than software like NOTUNG (*Bansal, Alm & Kellis, 2012*). Unfortunately, this model proved unsuitable as it too does not allow for incongruence events.

There are also several probabilistic reconciliation methods available (*Rasmussen & Kellis, 2007*, *2011*; *Ma et al., 2008*; *Doyon et al., 2010*; *Doyon, Hamel & Chauve, 2012*). While these models make use of more sophisticated models of evolution, they are far more computationally intensive and are only applicable to species for which speciation times and/or ancestral population size estimates are available, which is not the case for most species. PHYLDOG overcomes some of these limitations as it is able to estimate the most likely gene trees, species tree, and evolutionary history of a large number of

gene families at once (*Boussau et al., 2013*). Though it does not explicitly model ILS, the authors state that the algorithms can accommodate it as long as the signal is not too strong. This makes it unsuitable, as we expect gene families involved in direct environmental interactions to have a strong ILS signal. Further, this software is designed to combine the information from many gene families at once, and requires extremely significant computational resources (*Chaudhary et al., 2015*).

## Validation

A previous study conducted a detailed analysis of the vertebrate *cyp* gene family (*Thomas, 2007*), and found that enzymes with known xenobiotic substrates (about half of the gene family) exhibited high phylogenetic instability, while those with known endogenous substrates were strikingly phylogenetically stable, with clearly defined orthologous relationships. We validate the accuracy of MIPhy by comparing its predictions to the results of that study. That work relied upon the author's detailed knowledge of the gene family under study, and so was not quantified. As the genomes of an increasing number of species are being made available, manual analysis of large gene families from hundreds of species will become intractable. Further, it is desirable to use an algorithm that is consistent and deterministic.

Nematode collagens are a large multi-gene family of structural proteins. The *Caenorhabditis elegans* genome contains 181 collagen genes (*The C. elegans Sequencing Consortium, 1998*), many of which encode for proteins that form a major part of the nematode cuticle, which molts five times in the nematode life-cycle and protects the worm from environmental insult. A combination of high throughput and targeted gene knock-down studies have shown that 28 of these genes are associated with an observable phenotype, ranging from morphological variants to lethality (reviewed in (*Page & Johnstone, 2007*)). Available genome sequences from other *Caenorhabditis* species reveal both conservation and divergence of genes and their role in biochemical pathways (*Stein et al., 2003*; *Fierst et al., 2015*; *Gilabert et al., 2016*). To validate MIPhy's predictions for researchers aiming to prioritize genes for functional characterization, we test whether MIGs with lower phylogenetic instability scores were more likely to contain *C. elegans* genes associated with phenotypic changes when knocked-out.

While an individual can manually cluster a small tree without much trouble, the large size of some gene families and the ever-expanding availability of sequence data mean that this will quickly become intractable. There are several software packages used to automatically cluster a phylogenetic tree, but because of the ill-defined nature of clustering problems in general, the methods generally come to different conclusions on the same data sets. We are aware of no method that is targeted towards multi-species gene families, which means that none make use of problem-specific information such as an event-inference model of gene evolution. The clustering algorithm described here combines the similarity between each gene with the most parsimonious explanation of gene events, to predict the ancestry of each observed member of the gene family.

## METHODS

### Running MIPhy on a large phylogeny

The NCBI genome database (https://www.ncbi.nlm.nih.gov/assembly/organism/) was filtered for all animal genomes that were at a "Chromosome" or "Complete" level of assembly on July 26, 2016, yielding 98 hits. When there were multiple genome assemblies for a single species, only that with the highest number of annotated proteins was kept. Finally, the *Bos indicus, Capra aegagrus, Mus spretus, Nasalis larvatus,* and *Nomascus leucogenys* genomes were discarded as they were judged to contain too few protein sequences to have reliable annotations (all had fewer than 1,500). All protein sequences for the remaining 58 species were concatenated into one file, which was queried with the 628 vertebrate Cyp proteins from (*Thomas, 2007*) using blastp (*Camacho et al., 2009*), and resulting in 5,498 hits with an *E*-value $< 10^{-10}$. We note that this is not a particularly rigorous procedure; some of these sequences may not actually be Cyp proteins, and we may have missed some true hits. However, the purpose of this procedure was to generate a very large and representative phylogeny as a test case for MIPhy, not to comment on animal Cyps themselves.

The sequences were aligned using Clustal Omega (*Sievers et al., 2011*) with the command:

```
clustalo -i INPUT_FILE.fa –threads 10 –log INPUT_FILE-clustalO.log -v –force –use-kimura –iter 10 -o OUT_FILE
```

The "use-kimura" option specifies that a correction should be applied to the distance between sequences to better estimate their true evolutionary distance (*Kimura, 1980*). The columns of this alignment with <75% gaps were used to build a phylogenetic tree using RAxML (*Stamatakis, 2014*) with the command:

```
raxml -s INPUT_FILE.phylip -T 10 -# 5 -m PROTGAMMAWAG -j -p 12345 -n OUT_FILE
```

Here, the "-T" option specifies the number of threads used, "-#" specifies the number of iterations, and "-p" is just a random number seed to allow reproduction of the results. The "PROTGAMMAWAG" model was chosen, which uses the empirical amino acid frequencies and fits a gamma model of rate heterogeneity onto the LG substitution model.

### Analysis of nematode collagen genes

From Wormbase (*Howe et al., 2016*), there are 157 genes from *C. elegans* annotated with the gene class "col." To these we added the 19 genes listed in (*Page & Johnstone, 2007*). A further five were found by searching for the repetitive Gly-X-Y amino acid motif and checking each entry in WormBase. Phenotype data from gene knock-down studies is available from Wormbase. The protein sequences of *C. angaria, C. brenneri, C. briggsae, C. japonica, C. remanei, C. sinica*, and *C. tropicalis* were downloaded from Wormbase (version WS259). We searched the 181 *C. elegans* collagens against these protein sets using BLASTP (*Camacho et al., 2009*) and confirmed the presence of the characteristic and
repetitive Gly-X-Y amino acid motif. In instances of different isoforms, we selected the longest for subsequent analysis. In total, 1,349 genes were collected from the eight species.

The diversity of the N- and C-terminus across the collagens, coupled to the variable number of the Gly-X-Y motif, precludes a standard sequence alignment based approach. Therefore, we constructed a distance matrix based on k-mer frequency, using the jD2Stat program (*Chan et al., 2014*) with the command:

```
java -Xmx20g -jar jD2Stat_1.0.jar -n 1 -k 8
```

Here, "-Xmx20g" indicates we allocated the program 20GB of RAM, "-k 8" indicates we are using the default k-mer size of 8, and "-n 1" indicates we allow one wildcard character when identifying those k-mers.

We used the neighbor program with default parameters from the phylip suite to reconstruct the phylogenetic tree (*Felsenstein, 1989*). The species phylogenetic relationships had been previously determined using the ITS-2 genetic barcode (*Félix, Braendle & Cutter, 2014*). Note that *C. sp. 5* has since been renamed as *C. sinica* (*Huang et al., 2014*).

When statistically evaluating the instability scores between MIGs with and without observable knock-down phenotypes in *C. elegans*, neither set was normally distributed (via the Shapiro–Wilk test). We therefore used a one-tail Mann–Whitney *U* test to compare them.

## Parsimony clustering of the gene tree using a model of gene family evolution

The algorithm described in this work uses a model of gene family evolution derived from the core reconciliation methods of NOTUNG (*Chen, Durand & Farach-Colton, 2000*; *Vernot et al., 2008*; *Stolzer et al., 2012*), with some modifications such as allowing incongruence throughout the tree. We do this as apparent incongruence may arise for many reasons: due to errors in sequencing or gene-finding, incompletely resolved branches in tree-building software, HGT, ILS, or it may be due to selective pressures acting on one or more species. Using our model, each internal node of the gene tree is classified as representing one gene event: duplication, speciation, or incongruence. Gene loss is also considered a gene event, and is quantified at duplication nodes. The algorithm is detailed in Article S1, but summarized here.

MIPhy was designed to identify members of a gene family under adaptive evolution, and so must also cluster the given gene tree into MIGs. This is necessary to isolate "unstable" genes from "stable" genes, and has the effect of assigning all genes from all species in one MIG the same phylogenetic instability score. This score is a function of the model of gene family evolution, and for a given MIG it quantifies all gene events at or below the most recent common ancestor of that group:

$$\text{score}(g) = \theta_D \cdot D(g) + \theta_I \cdot I(g) + \theta_L \cdot L(g) + \theta_P \cdot P(g),$$

where $D(g)$, $I(g)$, and $L(g)$ are the total duplications, incongruence, and loss events within the MIG, respectively; $P(g)$ is a measure of the "relative spread" of the MIG (how dissimilar the sequences are—set to 0 for this phase); and the $\theta$ values are the strictly positive weights applied to each event. Under this definition, the score can be interpreted as a measure of the incongruence experienced by a cluster of genes throughout their evolutionary history.

Every node in the gene tree is evaluated in a depth-first post-order traversal; if the node is a leaf a new MIG is defined as containing only that node. At each non-leaf node in the tree the score function is used to compare two possibilities: merging all of that node's descendants into a single MIG, versus allowing the existing MIG patterns to remain. Initially, while traveling from the leaves towards the root, the "merge" choice tends to be most parsimonious. This continually populates the MIG, the final boundaries of which are determined by the point that the "remain" option instead becomes most parsimonious. This is the initial clustering phase, and it generates a preliminary clustering pattern.

## Cluster refinement

This initial clustering pattern arises from the most parsimonious history of gene events required to reconcile the gene family phylogeny ($T_G$) with the species phylogeny ($T_S$). It indicates which groups of genes, under this model, for the given weights and while disregarding all branch lengths in $T_G$, most probably evolved from a single homologue in an ancestral species. This second phase of the algorithm refines these predictions by incorporating branch length information, specifically the pairwise distance information between the sequences. If a sequence in the gene tree is separated by an uncommonly large phylogenetic distance from its closest MIG, there should be a cost associated with the decision to include it in that MIG.

This is accomplished by the "relative spread" term $P(g)$ in the score function, which measures the spread within a cluster. It is a measure of how "good" a cluster is compared to the others:

$$P(g) = \frac{\sigma(g)}{\bar{\sigma}} - 1,$$

where $\sigma(g)$ is the standard deviation of the points representing the sequences in the MIG rooted by $g$, and $\bar{\sigma}$ is the median standard deviation of all MIGs (excluding singleton clusters). The spread quantity is normalized around 0, so $P(g) = 1.0$ indicates that the spread of MIG $g$ is 100% larger than the median spread, while $P(h) = -0.3$ indicates that the spread of MIG $h$ is 30% smaller than $\bar{\sigma}$. Many clustering metrics, including this one, can only be calculated from data in a coordinate space, and so we first transform the phylogenetic tree into a set of points using multi-dimensional scaling (*Torgerson, 1952*) (see Article S1 for implementation details). Standard deviation is used as a measure of the pairwise branch lengths within a MIG because it is widely used and intuitive, but clustering-specific methods like the Davies–Bouldin index (*Davies & Bouldin, 1979*) or silhouette (*Rousseeuw, 1987*) could be easily substituted. As in the initial clustering phase,

each node $g$ in $T_G$ is again visited in turn. The clustering procedure is repeated, this time using the full score function.

## RESULTS

### Program input, workflow, and interface

This software requires two input files: the gene tree in Newick format, and an information file that contains the species tree (topology only; no branch lengths) as well as the assignment of each sequence to one species. MIPhy is agnostic to the method used to generate the tree and can be used to analyze those produced from nucleotides, amino acids, or any other features. The cluster analysis algorithm is written in Python and a local daemon server is started along with an HTML document to display the results. This page has interactive controls and communicates directly with the Python server, allowing the user to reanalyze their data and see the effects of modifying any of the parameters in real time.

The visualization page displays the gene tree clustered into MIGs, the current parameter values, summary statistics, and a sortable list of the MIGs (Fig. 1). Selecting a specific sequence or MIG will provide additional details. The page also contains a usage description, and provides options to modify visual elements like font sizes, the tree size, and the color of each element. The tree and legend can be exported and saved as an SVG image file, or the clustering pattern and instability scores from one or more species can be exported and saved as a CSV file.

MIPhy was used to analyze a dataset of annotated vertebrate Cyp proteins, which consists of 628 sequences from 10 species (*Thomas, 2007*). The algorithm calculated the optimal clustering pattern in 0.2 s on a 2.7 GHz laptop. Loading the results in a web browser required ~5 s. Modifying parameter weights causes the clustering analysis to be rerun, and redrawing the new results is sped up as only a subset of the page elements need to be modified or recreated (<1 s). To determine how MIPhy will scale to cope with the ever-increasing number of genome sequences, we analyzed a tree of 5,498 Cyp protein sequences from 58 animal species. MIPhy completed the initial clustering phase in 30 s, the optional cluster refinement phase in 7 min, and loaded the results in a web browser in 1.5 min.

### Phylogenetic instability of human Cyp proteins

MIPhy was run with default parameters on the Cyp phylogenetic tree from (*Thomas, 2007*), and the 59 scores from human sequences were extracted and graphed (Fig. 2); the substrate classification, positive selection, and genome clustering results from the same study were overlaid. These scores fell into two broad categories: 31 were unstable with scores in the interval [18.2, 97.5], and 28 were stable with scores in [0.1, 10.9]. Of the stable sequences, 23 had low scores in [0.1, 5.7], and the remaining five had intermediate scores in [7.8, 10.9].

Among the MIGs with intermediate scores, Cyp-11B1 (steroid 11β-hydroxylase) and Cyp-11B2 (aldosterone synthase) appear to have been recently duplicated in the terrestrial vertebrates, and likely played a role in the ancient transition from sea to land

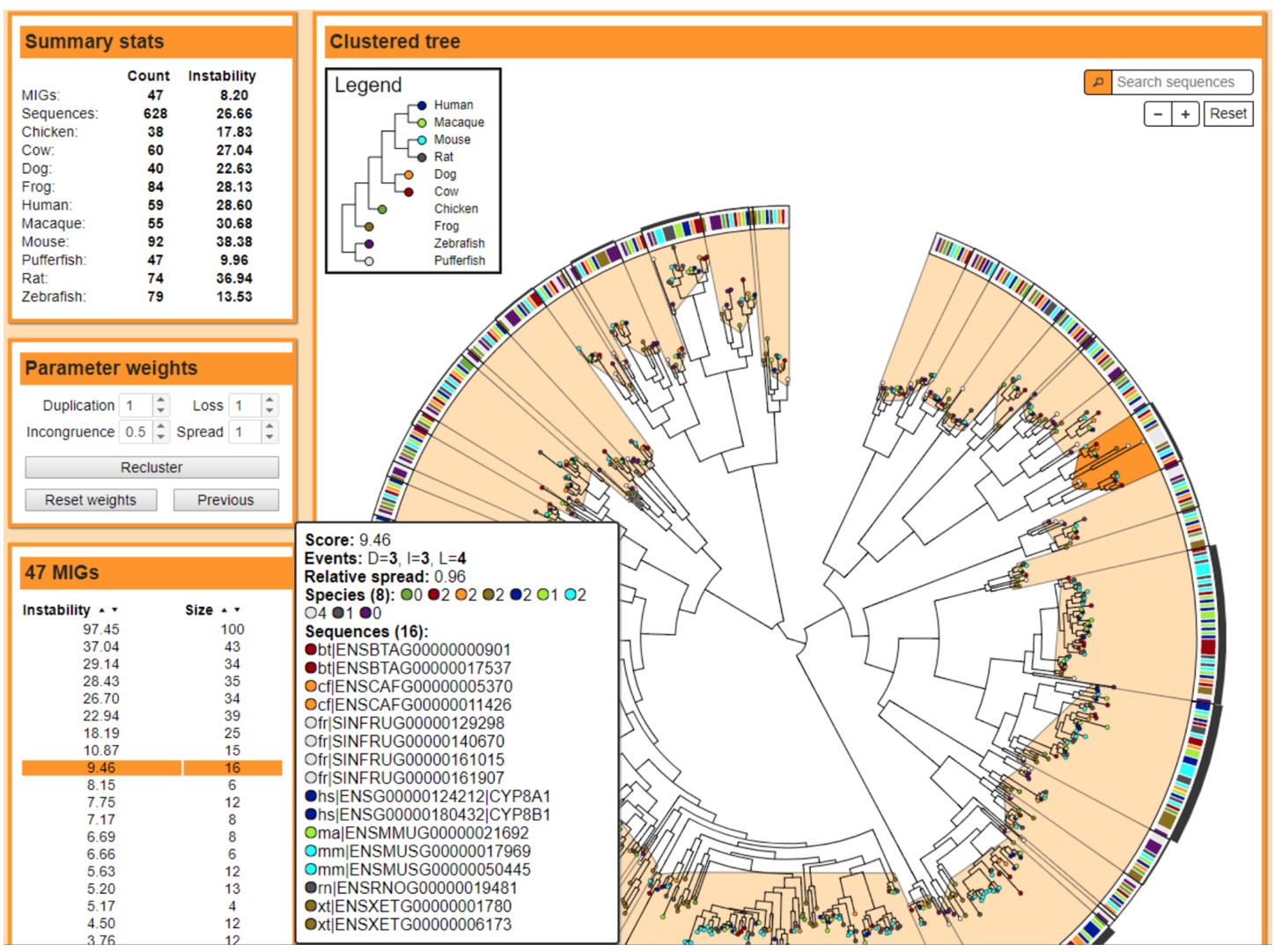

**Figure 1** **MIPhy results interface.** MIPhy visualization page for the 628 vertebrate Cyps from (*Thomas, 2007*). The MIGs are listed in the table on the left as well as indicated by the light orange shapes on the interior of the tree. The instability of each cluster is visualized by the bar charts around the outside of the tree. The colors of the band just inside of the circle match the colors of the tree nodes, and represent the originating species of each sequence.

(*Colombo et al., 2006*). Their instability score is elevated because rats appear to have two additional genes in that cluster and no homologs were found in chicken or frog. It is unclear whether they are actually lost in these species or simply absent from the assemblies.

## Parameter impact

The default MIPhy weight values are set at 1, 1, 0.5, and 1, for duplications, loss, incongruence, and spread, respectively. These have performed well in testing and analyses. The effects of modifying these values are considered in terms of the clustering pattern—which indicates which sequences are clustered together—and the cluster rankings—which indicates the instability score of each MIG relative to the others.

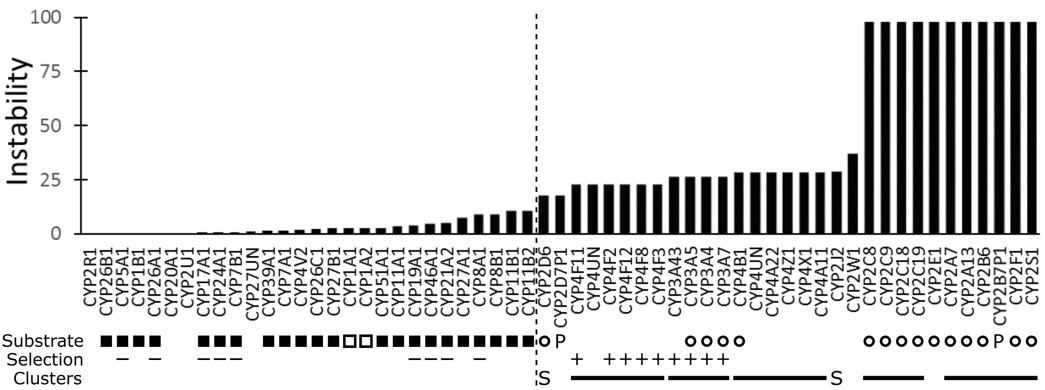

**Figure 2** **The phylogenetic instability of the 59 human Cyp proteins.** The vertical dashed line separates the stable from the unstable sequences. "Substrate" indicates those proteins with primarily endogenous roles (filled squares), primarily xenobiotic roles (empty circles), both xenobiotic and endogenous roles (empty squares), and pseudogenes (P). "Selection" indicates which of the 18 sequences tested showed evidence of positive selection (+), or no positive selection (−). In the "Clusters" row, the solid lines indicate those genes that are located in tandem arrays in the human genome or are syntenic with a tandem array in the mouse genome (S). All substrate, positive selection, and clustering data were taken from (*Thomas, 2007*).               

Increasing the weight for gene loss had very little effect; at even triple its default value it only caused four small MIGs out of the 47 from the vertebrate Cyp tree to be merged with their sister groups. Decreasing the gene duplication weight had much the same effect, causing five MIGs to be merged when it was set to 1/3 of the default value. Increasing the weights for duplication and loss together had no effect on the clustering pattern, and very minimal effect on the cluster rankings. Decreasing both weights together had the same effect as increasing the spread weight, which tended to break up larger MIGs. Decreasing the spread weight to zero had minimal impact, only merging two singleton groups with their neighbors. Decreasing the incongruence weight had no effect, and increasing it had little impact until it became very high, at which point it tended to break up groups.

## Phylogenetic instability of *Caenorhabditis* collagens

Across the eight species of *Caenorhabditis*, we found 1,349 collagen genes (Table S1). The characteristic Gly-X-Y repeat domain can vary greatly in length, presenting a problem for usual alignment guided phylogenetics. To overcome this, we used a k-mer based distance matrix (*Chan et al., 2014*). Default settings were used to cluster the protein phylogeny and subsequently score each cluster's phylogenetic instability (Fig. 3). A total of 244 MIGs were generated, with 41 MIGs containing proteins from all eight species, 60 MIGs covering any seven of the species, and 151 MIGs containing at least one protein from *C. elegans*. Twenty-five of the 151 MIGs that contained a *C. elegans* protein encoded by a gene whose knock-down is associated with an observable phenotype. The distribution of scores from these 25 MIGs was significantly smaller than the remaining 126 MIGs (medians = 2.02 and 3.22; *U* test statistic = 991; *p* = 0.002).

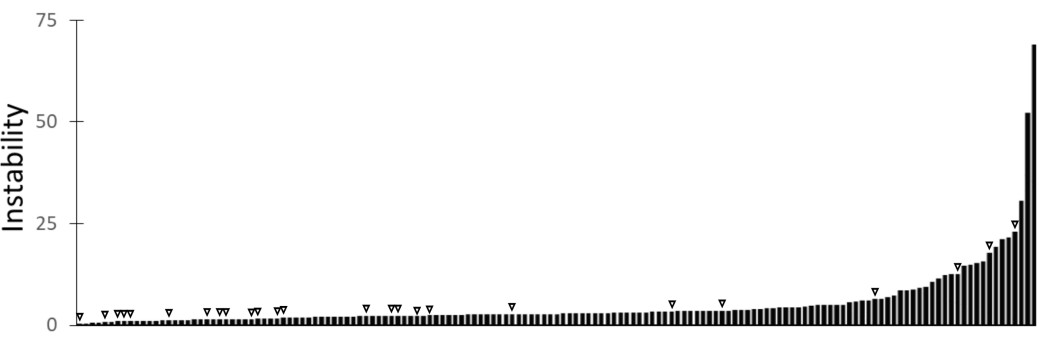

**Figure 3** **The phylogenetic instability of the 151 *C. elegans* collagen MIGs.** The MIGs containing genes with an observable knock-down phenotype are indicated by triangles.

## DISCUSSION

Positive selection, pseudogenization, and the presence of tandem gene arrays are characteristic of rapidly evolving genes, such as those involved in xenobiotic interactions (*Thomas, 2007*). Even though the MIPhy analysis does not incorporate any of this information, every human Cyp sequence with these characteristics received a high instability score (Fig. 2). These predictions appear to extend to the functional role of the enzymes as well, as MIPhy performed very well at classifying the human Cyp proteins into those primarily acting on xenobiotic or endogenous substrates. All enzymes with known endogenous functions had low scores, while all but two with primarily xenobiotic substrates had high instability scores; these exceptions were Cyp-1A1 and Cyp-1A2. While the latter is one of the most important human enzymes involved in xenobiotic metabolism, it has been suggested that both also have important endogenous roles (*Zhou et al., 2009*; *Kapitulnik & Gonzalez, 1993*), which may have shaped their evolutionary history in the vertebrate species studied here.

The predictions can be extended to species for which detailed substrate specificity information is limited. The sequences from terrestrial species in the MIG containing human Cyp-27A1 appear stable, but those of the aquatic or amphibious species do not. This observation suggests that these paralogs may play some role specific to aquatic environments. A similar observation can be made about the cluster containing human Cyp-2W1. It has the second-highest instability score, and of the 43 total sequences there is only one each from human, macaque, mouse, and cow. There are 16 from frog, 10 from zebrafish, and four from pufferfish, which would suggest that these paralogs may also have evolved to metabolize substrates specific to an aquatic environment, and that this capacity was lost in terrestrial species.

The collagens are a large gene family encoding for structural proteins. Most members have been investigated in the past using gene knock-down assays in *C. elegans*, resulting in observed phenotypic changes for approximately 15%. While not all the remaining genes have been investigated, many have, suggesting wide-spread functional redundancy. Using MIPhy to cluster and score the collagen gene phylogeny showed that we could prioritize genes for detailed functional assays, and that a low phylogenetic

instability score was a good predictor of genes with observable knock-down phenotypes. Further, this demonstrated that MIPhy is agnostic to the methods or underlying characters used to construct a gene tree, and so is applicable to a wide range of data.

An additional use of MIPhy is in the naming of genes, specifically towards generating hierarchical naming conventions using an evolutionary framework. Because a sequence identity threshold was used when annotating Cyp proteins, one may reasonably assume that Cyp-3A4 and Cyp-3A5 have related functions, as they are likely closely related. Conversely, no such assumptions may be made about many other gene families, whose members have often been annotated in order of discovery. This can pose a problem with the discovery of novel genes. If two species possess the example genes *pqr-21* and *pqr-22*, and one of them additionally possesses a paralog to *pqr-21*, this paralog will be named with the next available number; perhaps *pqr-42*. This single tiered naming system does not accommodate any way to suggest that *pqr-21* and *pqr-42* are related to each other. We propose that a phylogenetic analysis like MIPhy could be used to cluster such a gene family into sub-families, and that these clusters could be used to inform a multi-tiered naming system that is better able to accommodate newly discovered gene members. This is an issue that is going to arise more often as increasing numbers of species are being sequenced.

The predictions from these analyses would be complimentary to a between-genes positive selection analysis, which is the most commonly used measure of adaptive evolution. While a codon-based positive selection test measures the patterns of sequence variation, phylogenetic instability combines the relative sequence variation between species (from the cluster spread and incongruence events) with the most likely history of duplications and losses.

However, MIPhy does have its limitations. It is very sensitive to the given gene tree and does not currently incorporate any measures of uncertainty such as bootstrapping. We recognize that this information could be useful in an analysis of phylogenetic instability—for example by differentiating true gene events from those that may simply be phylogenetic artifacts—but leave this for a future version of the software. There are also exceptions to the assumption that phylogenetic instability is a hallmark of adaptive evolution; the most well-known may be the beta-globin genes that form part of hemoglobin. These genes exhibit sequence polymorphism within and between human populations, lineage-specific expansions and contractions in gene cluster size, and yet continue to play a very vital endogenous role (*Hill & Wainscoat, 1986*; *Opazo, Hoffmann & Storz, 2008*).

## CONCLUSION

This work presents, to our knowledge, the first algorithm for simultaneous reconciliation and clustering of large gene families. MIPhy's instability score has proven to be a valuable tool in identifying the members of gene families that exhibit characteristics of adaptive evolution, predicting collagens that play an important functional role in *C. elegans*, and agrees very well with the known substrate specificity of human Cyp enzymes. It is a useful tool to gain an understanding of the evolution of large gene

families, and to generate hypotheses about the potential functional roles of both the stable and unstable sequences.

## ACKNOWLEDGEMENTS

We thank everyone that has tested the software during its development. We also wish to thank Dr. Dannie Durand for her work on reconciliation algorithms, and for discussions on the nuances and applications of this work. Finally, we are grateful to Dr. James Thomas for providing us with his past data so that MIPhy could be validated on his published work.

### Funding

This work was supported by the Natural Sciences and Engineering Research Council of Canada (NSERC) through a Discovery Grant (#06239-2015) to James Wasmuth, a Collaborative Research and Training Experience Program (CREATE) program in Host-Parasite Interactions (#413888-2012) to James Wasmuth and John Gilleard (and others), and by Alberta Innovates—Technology Futures through a doctoral scholarship to David Curran. The funders had no role in study design, data collection and analysis, decision to publish, or preparation of the manuscript.

### Grant Disclosures

The following grant information was disclosed by the authors:
Natural Sciences and Engineering Research Council of Canada (NSERC): #06239-2015.
Collaborative Research and Training Experience Program (CREATE) program in Host-Parasite Interactions: #413888-2012.
Alberta Innovates—Technology Futures.

### Competing Interests

The authors declare that they have no competing interests.

### Author Contributions

- David M. Curran conceived and designed the experiments, performed the experiments, analyzed the data, contributed reagents/materials/analysis tools, prepared figures and/or tables, authored or reviewed drafts of the paper, approved the final draft.
- John S. Gilleard conceived and designed the experiments, authored or reviewed drafts of the paper, approved the final draft.
- James D. Wasmuth conceived and designed the experiments, analyzed the data, contributed reagents/materials/analysis tools, authored or reviewed drafts of the paper, approved the final draft.

### Data Availability
MIPhy is freely available at https://github.com/dave-the-scientist/miphy under a BSD 2-clause license. It is also available as an online tool at http://miphy.wasmuthlab.org.

## Supplemental Information

Supplemental information for this article can be found online at http://dx.doi.org/10.7717/peerj.4873#supplemental-information.

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
