# Peer review of "MIPhy: identify and quantify rapidly evolving members of large gene families"

_PeerJ, doi:10.7717/peerj.4873_

## Round 0.1 · original submission · Minor Revisions

Dear colleagues,

Thank you for the submission of this interesting article. As the reviewers point out, the manuscript is very well presented and very close to a final format.

Please make the clarification requested by reviewer #2 regarding borders of MIGs, and also correct the mistakes pointed out.

The reviewer also points out an issue with bootstrap assessment, which you address in a satisfactory manner (I will not request implementation of a whole feature in this version of the software). However, the reviewer has a good point which also made me concerned as a potential user. Would it be possible to perhaps comment a little more on how the bootstraps of the original input tree may affect the outcome? If both the species tree and the gene-tree are robust, the outcome should be robust as well (or am I wrong?). But then, how does the program behave if the species tree is robust but the gene tree not so much, or vice-versa?

Additionally, I have some comments of my own:

1. in methods section, you give command lines for several intermediate steps made using external software (clustal, etc). While command lines are always useful, I suggest actually explaining each parameter used (briefly). This will both reach a broader readership (people unconfortable with command lines), and also will make people confortable using other software for the same task (you may keep the command lines in there, but please explain also).

2. Figure 3 has indications of positive selection for specific sequences. I could not find out how you tested these sequences for positive seleciton. It may just be an outcome of the program (I unfortunately have not had time to test it), in which case, just please make this clear.

dan

Reviewer 1 ·

Basic reporting

Writing and organization were appropriate and professional throughout. Literature references were sufficient. Figures were adequate - Figure 1 was exceptional. Article was self-contained and generally presented well.

Experimental design

This article was basically a presentation of a new method for phylogenetic clustering of gene families, including method validation on 2 empirical datasets. The methodology was described in adequate detail and compared informally to other methods considering similar problems. No head-to-head method comparison was performed, which could have strengthened the results. Validation of the proposed method was not very formal, but mostly consisted of a description of how the proposed method 'correlated' with previous empirical results. The method validation analyses and discussion were adequate, but additional analyses - particularly more formal method evaluation using either empirical or simulated data - would strengthen the manuscript.

Validity of the findings

The proposed method was well-described. Validation of the method on empirical datasets was largely qualitative and would have benefitted from a quantitative assessment using benchmark empirical datasets or simulated data.

Additional comments

The proposed methodology will probably be welcome in the field and does advance the rather esoteric methodology of gene-species tree reconciliation. Method validation would have been strengthened by including a formal assessment on empirical data or simulated data (ideally both), but the informal method assessment presented appears adequate for a methods presentation manuscript.

·

Basic reporting

The article meets all standards set by the journal. I have no comments to the structure and text besides three really minor points.

Line 232: Tg and Ts should be explained in the main text not only in the supplement.
Line 321: What does the measure U stands for?
Line 350: “…we could predict those with observable knock-ow phenotypes.” Could they specify how? Do they expect these genes to have always lowed instability score as in the case of collagens? If yes, could they explain, why they take this assumption.

Experimental design

The design of the study is clear and easy to follow. The method is described well in the manuscript and attached supplement, however I would like to ask for one important clarification. My lack of understanding may result from my poor understanding of some principles of the statistics behind.

Matter of clarification: I think that I understand in general, how the events are assigned to nodes and how the score is calculated for a node g. What I do not understand is, how the borders of MIGs are defined. As this is critical for the whole paper, I would be happy if the authors could explain this more explicitly for readers less trained in in mat and statistics like me.

The only imperfection of the tool, which I am able to recognise, is the fact that it does not take into account the robustness of the tree branching (measured e.g. in bootstraps). I would imagine that this feature would be useful particularly for assessing the events of incongruences. At this point they assume (Article S1, line 17) that “…incongruence is more likely than a duplication event followed by several independent loss events, so the latter is not considered as possible history.“ However, the latter is indeed a possible history, which should be in my opinion preferred over incongruence if the branch support for the daughter node(s) below the node of incongruence (n4 node in Figure S1) would be high.

The executables and testing file are provided and downloadable. The online web tool seems to be working as described. The graphical interphase is nice and easy to operate. The calculations are fast as stated in the manuscript.

Validity of the findings

The paper describes a tool for defining a sensible units (MIG) in a tree of large gene family and for quantification of “speed” of evolution within such units. As the phylogenetic analyses of large gene families is often un-easy, having such tool is by all means a sensible idea. They provide analyses of two gene families, which apparently support the usefulness of the software. Future and other users will assess the value of the tooll in their targeted analyses. To me, the tool seems functional, the interface is well-arranged, interactive and easy to operate. However, I was testing it only on the testing data sets provided by authors. I would recommend using the software for analyses of large gene family phylogenies.

Additional comments

I propose minor revision but I feel it is actually very close to "accept".

---

## Round 0.2 · accepted · Accept

Dear colleagues,

Thank you for the corrections made to the original submission. I believe the manuscript is now more robust, clear and accessible to a wider audience. The issue regarding branch robustness raised by reviewer #2 remains unresolved, but clearly outside the scope of this communication. I consider it to be an interesting avenue for further development of the software.

Thanks again and congratulations.

Kind regards,

dan

#